# Antioxidant Activity, Metabolism, and Bioavailability of Polyphenols in the Diet of Animals

**DOI:** 10.3390/antiox12061141

**Published:** 2023-05-23

**Authors:** Drago Bešlo, Nataša Golubić, Vesna Rastija, Dejan Agić, Maja Karnaš, Domagoj Šubarić, Bono Lučić

**Affiliations:** 1Faculty of Agrobiotechnical Sciences Osijek, J. J. Strossmayer University Osijek, Vladimira Preloga 1, HR-31000 Osijek, Croatia; golubicnatasa@gmail.com (N.G.); vrastija@fazos.hr (V.R.); dagic@fazos.hr (D.A.); mkarnas@fazos.hr (M.K.); dsubaric@fazos.hr (D.Š.); 2NMR Center, Ruđer Bošković Institute, Bijenička cesta 54, HR-10000 Zagreb, Croatia; lucic@irb.hr

**Keywords:** animal nutrition, supplementation of polyphenols, bioavailability of polyphenols, biotransformation, metabolism, antioxidant/pro-oxidant activity, diversity of microbiota, gut microbiota composition, immunomodulation, animal health, animal product quality

## Abstract

As the world’s population grows, so does the need for more and more animal feed. In 2006, the EU banned the use of antibiotics and other chemicals in order to reduce chemical residues in food consumed by humans. It is well known that oxidative stress and inflammatory processes must be combated to achieve higher productivity. The adverse effects of the use of pharmaceuticals and other synthetic compounds on animal health and product quality and safety have increased interest in phytocompounds. With the use of plant polyphenols in animal nutrition, they are gaining more attention as a supplement to animal feed. Livestock feeding based on a sustainable, environmentally friendly approach (clean, safe, and green agriculture) would also be a win–win for farmers and society. There is an increasing interest in producing healthier products of animal origin with a higher ratio of polyunsaturated fatty acids (PUFAs) to saturated fatty acids by modulating animal nutrition. Secondary plant metabolites (polyphenols) are essential chemical compounds for plant physiology as they are involved in various functions such as growth, pigmentation, and resistance to pathogenic organisms. Polyphenols are exogenous antioxidants that act as one of the first lines of cell defense. Therefore, the discoveries on the intracellular antioxidant activity of polyphenols as a plant supplement have contributed significantly to the improvement of antioxidant activity, as polyphenols prevent oxidative stress damage and eliminate excessively produced free radicals. To achieve animal welfare, reduce stress and the need for medicines, and increase the quality of food of animal origin, the addition of polyphenols to research and breeding can be practised in part with a free-choice approach to animal nutrition.

## 1. Introduction

Plant-based dietary supplements for domestic animals have increased significantly in the last decade. Polyphenols are secondary metabolites of plants that contain bioactive compounds and have beneficial effects on animal organisms. Polyphenols in plants play an important role in growth and reproduction and provide protection against pathogens and herbivores [1]. There is great interest in these phytochemicals for their health benefits and effects on animals [2,3]. Phenolic compounds, also known as polyphenols, are a class of compounds found in various plant species. They are built from one or more aromatic rings on which one or two hydroxyl groups are present. There are three main groups of polyphenols: flavonoids, non-flavonoids, and tannins. Their biological function depends mainly on the chemical structure [4]. More than 10,000 compounds have been identified that have anti-inflammatory, immunomodulatory, and antimutagenic properties. The positive effects of phenolic compounds are attributed to their antioxidant activity [5]. Their role as antioxidants is comparable to important biological and well-known antioxidants: vitamins E and C. Despite these benefits, polyphenols are characterized as substances with low bioavailability, and additional research is needed to investigate their efficacy in the diet of domestic animals [2]. The higher level of unsaturated fatty acids compared to saturated fatty acids arouses interest in the production of healthier animal products in order to change the diet of animals. This dietary strategy has been linked to increased lipid peroxidation, a process in which free radicals “steal” electrons from lipids in cell membranes, leading to cell damage. It is important to maintain the quality of meat and dairy products by mitigating oxidative decay. The addition of antioxidant molecules to food or to the final product controls and reduces the occurrence of oxidative stress.

It is extremely important to understand and relate the inflammatory processes in animals to prevent oxidative stress. Oxidative stress is the result of an imbalance between reactive oxygen species (ROS) and antioxidants. These include enzymatic antioxidants (e.g., carotenoids, tocopherols, polyphenols, and glutathione) and antioxidant enzymes (e.g., superoxide dismutase (SOD), catalase (CAT), and glutathione peroxidase (GPX)) [6]. The accumulation of ROS in cells can seriously damage macromolecules and continuously (in a chain) stimulate the production of ROS.

Oxidative stress contributes to the development of various diseases and chronic pathological conditions. It is known to be involved in the development of pneumonia [7] and sepsis [8] in pigs. It also occurs, for example, in piglets that are switched from mother’s milk to solid dry feed [9], because the digestive tract and the body’s own system are not yet mature, resulting in reduced appetite and stunted growth [10]. In animals exposed to stressful conditions such as transport, starvation, and low and high temperatures, glucocorticoids (CTC) are released, leading to an increase in free radicals and an increased rate of oxidation of microfibrillar proteins, resulting in the inhibition of muscle growth and muscle wasting [11,12].

Antioxidant compounds are needed to prevent the formation of new free radicals, and polyphenols are prominent compounds that react with free radicals. The formation of free radicals ultimately leads to oxidative stress, and the addition of antioxidant compounds can stop the further spread of ROS. In addition, some experimental studies have confirmed positive results in vivo with the addition of polyphenols, indicating their potential as natural antioxidants [13,14,15].

Interest in the use of natural antioxidants in food production instead of synthetic antioxidants has increased in recent years, as they are less harmful to the environment and are also used for economic reasons. In addition, natural antioxidants are better for the end consumer as they are considered safer [4].

Recently, there has been growing interest in the use of by-products in animal nutrition because they contain high levels of unsaturated fatty acids (PUFA) as well as a high concentration of polyphenols [16]. This diet may contribute to a greater stability of poultry meat to fatty acid oxidation and a greater stability of meat products for human consumption [17,18]. In pig diets, the use of grape pomace (GP) with a high PUFA content (60.9–64.4%) and a high PUFA/SFA ratio (2.80–3.0) resulted in improved effective growth, altered the composition of fatty acids in adipose tissue, and resulted in better meat quality [19,20]. 

A review of the literature on the supplementation of polyphenols in animal diets was done to follow the changes (metabolism and biotransformations) of polyphenols in the digestive tract of monogastric animals. Some monogastric animals may consume foods of animal origin in addition to a plant-based diet, such as pigs, dogs or birds, while horses or rabbits consume plant-based diets. Results from the literature on the bioavailability, biotransformation and metabolism of polyphenols in relation to the gut microbiota and its diversity are considered and reviewed. Biological activities such as antioxidant and pro-oxidant activities are analysed as well as the effects of polyphenols on animal growth and the immune system by modulating the diversity of the gut microbiota in animals. Finally, the results of polyphenol supplementation on animal health and the quality of animal products used in human nutrition will be analysed.

## 2. Classification of Polyphenols

The classification of polyphenols is based on the number of phenolic rings and the structural elements that connect the rings. They are divided into four classes: phenolic acids, flavonoids, stilbenes, and lignans (Figure 1). The phenolic acids are further subdivided into hydroxybenzoic acid and hydroxycinnamic acid. About one-third of the polyphenolic compounds are phenolic acids and are found in almost all plant materials. Caffeic acid, gallic acid, and ferulic acid are some of the most common phenolic acids.

The most common polyphenols are flavonoids, which have a common basic structure. They consist of two aromatic rings joined by three carbon atoms, forming an oxygenated heterocycle.

Flavonoids have been classified into different subclasses based on the substitution content of the C ring, the oxidation state of the heterocyclic ring, and the position of the B ring (Figure 2). For these reasons, flavonoids are divided into seven main subclasses: flavanols, flavonones, isoflavones, anthocyanins, chalcones, flavones, and flavonols (Figure 2). In flavanones, flavones, flavonols, flavanols, and anthocyanins ring B is located at position 2 of the heterocyclic ring, while in isoflavonoids at position 3. Flavanones and flavonols have a saturated central heterocyclic ring and, in this case, one or more chiral centers are present. On the other hand, anthocyanins, isoflavones, flavones, and flavonols have an unsaturated central heterocyclic ring and the molecule is achiral [22,23].

Stilbenes contain two phenyl residues linked by a two-carbon methylene bridge. Most stilbenes in plants act as antifungal compounds that are synthesized only in response to infection or injury. The most extensively studied stilbene is resveratrol. Lignans are diphenolic compounds with the structure of 2,3-dibenzylbutane formed by the dimerization of two cinnamic acid residues [21,24]. 

## 3. Reactive Oxygen Species and Reactive Nitrogen Species

All aerobic organisms have a need for molecular oxygen, and at the same time it is ironic that they have to defend themselves against the dangerous oxidase [25]. The basis of aerobic life is the oxidation of organic compounds, namely carbohydrates, proteins, and fats, which provide the metabolic energy necessary for life functions. Free radicals and antioxidants have become common topics in today’s discussion of disease mechanisms [26]. It is normal for various metabolic processes to produce free radicals. The formation of free radicals is well regulated by the physiological process in the aerobic cell. In homeostasis, the formation of free radicals and their elimination are in equilibrium because during the formation of free radicals in the cell, there is an antioxidant defense system that eliminates ROS and reactive nitrogen radicals (RNS, reactive nitrogen species). When an imbalance occurs, oxidative stress results [27].

### 3.1. Sources of ROS

The production of ROS is a natural component of aerobic life, responsible for various cellular functions, from oxygen transport pathways to defense against microbial invasion and gene expression to growth and death promotion [28,29,30,31]. ROS can be produced in the mitochondria, where O_2_ is reduced to O_2_·^−^ in the ETC (electron transport chain) process, where the transfer of electrons from NADH and FADH_2_ to O_2_ can lead to a “leakage” of electrons at complexes I and III. However, the enzymes monoamine oxidase, 2-ketoglutarate dehydrogenase, and glycerol phosphate dehydrogenase, which are located in the mitochondrion, can additionally contribute to production [32,33,34].

A reaction takes place at the inner membrane of the mitochondria in the respiratory chain in which partial intermediates O_2_ can be formed [35].
$$O_{2}\overset{+e^{-}}{\to }\cdot O_{2}^{-}\overset{+e^{-}\left(+2H^{+}\right)}{\to }H_{2}O_{2}\overset{+e^{-}}{\to }\cdot \mathrm{HO}+{\mathrm{OH}}^{-}\overset{+e^{-}\left(+2H^{+}\right)}{\to }2H_{2}O$$

The three primary species, i.e., the superoxide anion (O_2_^−^), the hydrogen peroxide (H_2_O_2_), and the hydroxyl radical (·OH), are called oxygen with reactive properties. O_2_^−^ and ·OH are usually referred to as “free radicals” and H_2_O_2_ and ^1^O_2_ as non-radicals. An excessive amount of ROS is very toxic to the cell. A usually high concentration of ROS in a cell is called “oxidative stress” and can damage various molecules in the cell, e.g., lipids, proteins and DNA. In particular, in the cell membrane, it can act as a double bond and cause lipid peroxidation, increasing the permeability and fluidity of the membrane. In damaging proteins ROS can cause site-specific modifications, fragmentation of protein chain, as well as susceptibility to proteolysis [36]. Finally, ROS can damage DNA through deoxyribose oxidation, strand scission, nucleotide removal, base modification and DNA–protein cross-linking [13,14,15,37]. The protective effect against ROS is exerted by several enzymes, namely superoxide dismutase (SOD), catalase (CAT) and glutathione peroxidase (GPx), as well as non-enzymatic compounds such as vitamin E, β-carotene, ascorbic acid, glutathione (GSH) and exogenously supplied polyphenols [38,39,40,41].

In addition to the production of ROS in organelles, several enzymes, including cytochrome P450 (CYP) 2E1, NADPH oxidase (NOX), cyclooxygenases, xanthine oxidases, and lipoxygenases, produce ROS in the plasma membrane and cytosol [42].

Similarly, sources of the formation of oxidants are unwanted substances in food, such as pesticides, organic solvents or mycotoxins. These compounds induce the xenobiotic system of the liver, which produces oxidants as by-products [43]. Oxidative stress is directly related to inflammation because oxidants activate NF-kB, an important regulator of inflammation [44]. NF-kB is a protein complex found in almost all types of animal cells and i bound in an inactive state to inhibitory proteins in the cytosol. Upon stimulation by oxidants and various other stimuli such as cytokines, free radicals, heavy metals, bacterial stimuli and viruses, inhibitory proteins are released from nuclear factor kappa B (NF-kB), facilitating the translocation of active NF-kB into the nucleus and activation of transcription of a large number of genes involved in all aspects of inflammation (e.g., vasodilation, chemotaxis, cell adhesion and phagocytosis) [45]. Typical proteins encoded by NF-kB target genes include pro-inflammatory cytokines, chemokines, inflammatory enzymes, adhesion molecules and various receptors. Several NF-kB-regulated proteins such as cytokines and chemokines stimulate the production of oxidants by activated neutrophils (respiratory burst) and their mitochondria, thereby promoting oxidative stress [46]. If this cycle in which oxidative stress develops cannot be broken due to long-term and excessive production of oxidants, the inflammatory process becomes chronic and cells and tissues are damaged [47].

### 3.2. Nitrosative Stress

RNS are nitric oxide (NO·), peroxynitrite (ONOO·) and nitric oxide NO_2_ [42,48]. NO is formed from L-arginine by three major isoforms of nitric oxide synthase (NOS). It is a very versatile molecule with numerous functions and mechanisms of action. It is described as a diffusing radical that leads to vasodilation and plays a key role in the vascular system. Post-translational modifications mediated by NO lead to the formation of another reactive nitrogen with superoxide, resulting in peroxynitrate ONOO· [49]. In an aqueous environment, the peroxynitrous acid HONO is formed, which can dissociate with NO_2_ and the highly reactive hydroxyl radical (·OH) [50,51]. The hydroxyl radical is reactive enough to remove electrons from almost any biological molecule. For example, it forms a tyrosyl radical with tyrosine, which can react with NO_2_ to form nitrotyrosine.

The phagocyte immune system uses the phagocyte’s ability to modify proteins and other molecules. By swallowing and killing bacteria due to the activation of macrophages and neutrophils, ROS are released in a “burst of breath”. After phagocytosis of antigenic particles, NADPH oxidase produces large amounts of superoxide [52,53,54,55,56,57]. Furthermore, the bactericidal function of activated macrophages depends on the presence of peroxynitrite and the fact that inhibition of superoxide or nitric oxide generation inhibits this function [54]. While peroxynitrite serves to protect the body from pathogens, its overproduction or dysregulation of these metabolic pathways is also harmful.

### 3.3. Free Radicals and Internal Defense

Oxidative stress means a disturbance in the balance between the formation of free radicals and the ability of the body’s defense systems to eliminate them [58]. Oxidants are free radicals, which are atoms, ions or molecules that contain an unpaired electron in the outer electron shell. The most common are reactive oxygen radicals (ROS) and reactive nitrogen radicals (RNS). ROS are superoxide radicals (O_2_^−^), hydroxyl radicals (OH), peroxyl radicals (RO_2_), alkyl radicals (RO), hypochlorous acid (HClO), ozone (O_3_), singlet oxygen (^1^ΔgO_2_) and hydrogen peroxide (H_2_O_2_). The hydroxyl radical is the most reactive, i.e., it is characterized by low substrate specificity and a short half-life. Figuratively speaking, it can most easily split an electron from the surrounding molecules. It is an important trigger of lipid peroxidation.

RNS are reactive species that contain both nitrogen and oxygen, e.g., nitric oxide (NO), nitrogen dioxide (NO_2_), peroxynitrite (ONOO^−^), etc. NO produces various isoforms of the enzyme NOS (nitric oxide synthase) [42]. At low concentrations, it is essential as a neurotransmitter and hormone that causes vasodilation. In high concentrations, it combines with an oxygen molecule or a superoxide radical to produce RNS, whose effect on cells is similar to that of ROS [59,60].

There are physiological mechanisms in cells to suppress ROS when they become out of control under conditions of oxidative stress. These negative effects can be significantly attenuated by antioxidants, a heterogeneous class of chemicals whose common feature is the ability to interrupt radical chain reactions and thus prevent or limit cell damage. In cells, there are defense mechanisms that attempt to maintain homeostasis. These include the enzymes superoxide dismutase (SOD), catalase, and glutathione peroxidase.

Superoxide dismutase catalyzes the following reaction:O_2_· + O_2_· + 2H^+^ → H_2_O_2_ + O_2_

Catalase:2H_2_O_2_ → 2 H_2_O + O_2_
ROOH + AH_2_ → H_2_O + ROH + A

Glutathione peroxidase:ROOH + 2GSH → ROH + GSSG + H_2_O 

### 3.4. Exogenous Molecules in Defense against Oxidative Stress

Plants are a tremendous source of antioxidants. Many of them are already known and widely used, and many others are still waiting to be discovered dscovered. Plant antioxidants generally have the chemical character of polyphenols and can break radical chain reactions by forming phenoxy radicals, which are generally less reactive and/or converted into relatively stable dimers, quinones, etc. For a number of natural polyphenols, some studies on the structure-activity relationship are available [61,62,63,64], in which it was shown that the number and arrangement of OH groups in the structure of polyphenols are the most important factors for their antioxidant activity. In general, these arrangements of the OH groups reflect the dissociation energy of the OH bonds through which the polyphenols exert their activity as antioxidants.

Due to their proven therapeutic importance in prevention and treatment, bioactive molecules of plant origin have attracted much attention, whether whole plants, plant extracts, or even isolated compounds are used. Larger randomized studies are needed to provide clear evidence of the benefits/risks of antioxidant supplementation. Antioxidants are also sensitive to oxidation, so their use as foods (or food supplements) should be carefully considered as oxidation and reduction reactions do not take place in them in isolation. Taking high doses of antioxidants is coming under increasing criticism, as evidence of some adverse effects is accumulating. The investigation of their chemical constituents as future prophylactic and therapeutic agents is of particular interest as they are more effective and safer than those commonly available [65].

The cause of cell damage can be lipid peroxidation induced by free radicals, DNA chain cleavage, and protein oxidation [66,67,68]. ROS and RNS have important physiological effects in certain concentrations. It is known that living organisms can regulate the concentration of ROS and RNS through dietary intake of antioxidants and endogenous production of antioxidants and, for example, systems for inactivating excess radicals in order to maintain a balance between antioxidants and ROS/RNS that allows the body to function normally. An imbalance that favours the accumulation of ROS/RNS is defined as oxidative/nitrosative stress. Current indications suggest that oxidative/nitrosative stress is involved in several diseases, including liver disease [69,70].

Free radicals include not only ROS and RNS but also other radicals [71,72]—details are summarized in Table 1.

The enzymatic antioxidant defense system includes antioxidant enzymes such as superoxide dismutase, glutathione reductase, catalase, and other enzymes that play a key role in detoxifying radicals into non-reactive molecules [69,70] (Figure 3). Under physiological conditions, these enzymes keep the radical concentration in the cell low and their activity is regulated by precise mechanisms at the molecular level. All these enzymes are essential for maintaining homeostasis between oxidation and antioxidant capacity and for the survival of all aerobic organisms.

## 4. Distribution of Polyphenols in Nature

Polyphenols are found in almost all plants, They have been confirmed to be present in their various parts such as roots, leaves, flowers, fruits and seeds, which, among other things, protect plants from various pests and UV radiation [73]. Their distribution is variable at the tissue level. Thus, the outer layers of plants contain higher amounts of polyphenols than the inner layers, or the insoluble polyphenolic compounds are associated with the cell wall, while the soluble ones are found in the vacuoles [74]. In particular, prolonged scalding of the fruits of the plant, decreases the concentration of polyphenols decreases, especially at higher temperatures, which is a consequence of the sensitivity of polyphenols to oxidation [21]. Fruits, vegetables, legumes, nuts, and herbs are typical sources rich in polyphenols, and these foods are also consumed by animals [75,76]. Antioxidant properties are attributed to polyphenols, because they can act as chain breakers or radical scavengers, depending on their chemical structure [77,78].

Plants synthesize an incredible variety of metabolites with different properties. Secondary plant metabolites are generally associated with plant defense, but also with a variety of biological properties. The concentration of secondary metabolites and their activities in the biological system vary depending on the maturity of the plant and its parts, as well as the soil conditions, the availability of water and light, and other environmental conditions under which the plant grows [79].

In order to ensure the desired high productivity of livestock production, it is necessary to control oxidative stress and inflammatory processes, which are closely linked. Oxidative stress has become a major challenge in animal husbandry because it affects the growth of animals. Oxidative stress reduces the globulin concentration in plasma and thus reduces the immune status of poultry [80]. Many studies have been conducted to solve this problem and one of the safest solutions is the use of polyphenols. Polyphenols are well known as exogenous antioxidants that act as one of the first lines of cellular defense. The discovery of the intracellular antioxidant activity of polyphenols as a plant food supplement contributes significantly to the improvement of antioxidant activity because polyphenols prevent oxidative stress damage and eliminate excessively produced free radicals [80].

## 5. Digestion, Absorption, and Metabolism of Polyphenols

The biological effects of certain compounds contained in plant foods for animals or other plant preparations depend on their uptake, metabolism, distribution, and excretion from the organism, i.e., on the bioavailability after their uptake into the organism as well as on the reducing properties of the resulting metabolites. Understanding the processes involved in the uptake and distribution of polyphenols is crucial for determining their potential bioactive effects in vivo as well as their overall importance in the prevention of a number of diseases associated with oxidative stress [81,82,83,84].

### 5.1. Digestion/Metabolism/Biotransformation and Bioavailability of Polyphenols in Animals

Recent studies on the bioabsorption of polyphenols indicate their low bioavailability after after intake of relatively high doses. The main obstacle to their pharmacological use is their poor bioavailability, which is related to the interactions of polyphenols in the different phases of digestion, absorption, and distribution that alter their molecular structure, especially the interactions with food, digestive enzymes, and transporters in the intestine and blood proteins [85].

In explaining the biological effects of polyphenols, it was assumed that they are bioavailable and can effectively reach the target tissue. It is very important to understand the processes by which they are absorbed, metabolized, and excreted from the body. The study on absorption is complicated because of the molecular complexity of foods rich in polyphenols and other factors such as the degree of polymerization and conjugation with other compounds and phenols. Most polyphenols are present in food in the form of esters, glycosides or polymers, and cannot be absorbed in these forms. Once absorbed, polyphenols are recognized by the body as xenobiotics, so their bioavailability is relatively low compared to micro- and macronutrients.

The metabolism of polyphenols is similar to metabolic detoxification to reduce their potential cytotoxic effects by increasing their hydrophilicity and facilitating excretion in urine or bile [23]. The structure of polyphenols, not their concentration, determines the rate and also the extent of absorption as well as the type of circulating metabolites in plasma. Depending on their degree of structural complexity and polymerization, these compounds can be readily absorbed in the small intestine (monomeric and dimeric polyphenols) or almost unchanged reach the large intestine (oligomeric and polymeric polyphenols) [86].

According to some estimates, only 5–10% of ingested polyphenols can be absorbed in the small intestine. After absorption, less complex polyphenolic compounds can be hydrolyzed and biotransformed in enterocytes and then in hepatocytes. The result is a series of hydrophilic conjugated metabolites (methyl, glucuronide, and sulphate derivatives) that rapidly enter the bloodstream and are further distributed to organs or excreted in the urine [24]. It is difficult to track the fate of each compound in the body because ten new compounds may be formed when each phenolic compound is metabolised [87]. Very often it is not possible to detect the parent phenol, because in most cases metabolic processes convert phenolic antioxidants into completely different molecules [87]. As xenobiotics, they may first undergo oxidation, reduction or hydrolysis reactions (phase I metabolism) in enterocytes, introducing or exposing a functional group, such as a hydroxyl group, especially for conjugation (phase II metabolism) [87,88]. The aromatic structures in enterocytes usually remain intact, but the hydroxyl groups on the phenolic aromatic ring are successfully conjugated to glucuronide, sulphate, and/or methylated metabolites by the action of uridine-5-diphosphate glucuronosyltransferase (UGT), sulphotransferase (SULT) and catechol-*O*-methyltransferase. (COMT) (Figure 4) [89,90,91].

The conjugation of catechol antioxidants usually occurs mainly at the meta position [84,92,93]. However, conjugation with a glucuronic acid or sulphate group can also occur at the para position, usually following conjugation at the meta position with a methyl group.

### 5.2. Biotransformation of Polyphenols by the Gut/Colon Microbiota

#### 5.2.1. Fate of Polyphenols in the Stomach and Small Intestine

When food passes from the oral cavity to the esophagus, the ingested polyphenols enter the stomach. Due to low pH and contact with bacteria, some flavonoids can be degraded into phenolic acids, but in general no significant changes occur under the acidic conditions of the stomach [68]. Flavonoid glycosides can hydrolyze under acidic conditions in the stomach and pass unchanged into the small intestine.

Some of the metabolites from the enterocytes enter the duodenum via bile and are then hydrolyzed in the large intestine by bacterial enzymes (mainly β-glucuronidase). 95% of the body’s microbiota is found in the intestines, primarily in the colon, where microbial cells outnumber the total number of cells in the body [4]. With the sequencing methods available today, it is possible to determine the presence of a healthy or disturbed microbiome [86]. The intestinal microbiota of the digestive system plays an important role in health and disease, but our understanding of the composition, dynamics, and functionality of the intestinal ecosystem is still rudimentary [87].

#### 5.2.2. Polyphenols and the Gut Microbiome

This enterohepatic recycling can result in a longer retention of polyphenols in the body. The remaining polyphenols (90–95% of the total ingested polyphenols) may accumulate in the lumen of the large intestine, from where they are excreted under the action of enzymes together with conjugates into the lumen of the intestine. The intestinal microbial community produces metabolites such as aromatic acids (hydroxyphenyloctene, phenylpropionic acid, phenylbutyric acid, phenylvalerolactones, etc.) [86,94]. All these phenolic metabolites produced by microbes can be absorbed or excreted in the feces. Once absorbed, they enter the liver via the portal vein, where they can undergo extensive metabolism (including glucuronidation, methylation, sulphation, or a combination thereof) until they finally enter the systemic circulation and are distributed to organs or excreted in the urine. The intestinal microbiota is responsible for the extensive degradation of the original structures of polyphenols into a series of molecules of low molecular weight, which, because they can be absorbed, may be responsible for the biological activity resulting from the consumption of foods rich in polyphenols, rather than the original compounds in the food. After consumption, the concentration of polyphenols that is achieved varies considerably depending on the type of polyphenols and the food source. With normal consumption, the concentration of intact flavonoids in plasma rarely exceeds 1 µM, and peak concentrations of 1 µM are usually reached 1–2 h after ingestion [86]. Polyphenols and their derivatives are mainly excreted through urine and bile.

There are very few studies on the interaction of polyphenolics with the gut microbiota in animal diets. *In vivo,* studies have shown that resveratrol has great potential as an antibiotic alternative to reverse the negative effects of weaning stress in piglets infected with *E. coli* and *Salmonella* [95,96,97,98] on their growth, immunity, and microbiota [99]. Similarly, grape pomace supplementation in the diet of pigs reduces diarrhea caused by *E. coli* [100]. Fiesel et al. [100] showed that feeding pomace flour extract and grape pomace alters the microbial composition and leads to a reduction in *Streptococcus* spp. and *Clostridium* in the fecal microbiota.

#### 5.2.3. Reaction Phases of the Biotransformation of Polyphenols

The biotransformation reactions that take in place during phase I metabolism are oxidation, reduction, and hydrolysis. The main purpose of these reactions is to increase the polarity of heterogeneous phenolic compounds to facilitate their excretion [101,102]. Day et al. [91] found that these reactions significantly affect the antioxidant activity of flavonoids and their cross-reaction with proteins.

The second stage of biotransformation involves the incorporation of various chemical radicals into exogenous compounds. The free radicals transported in the body originate from endogenous, polar, and easily accessible molecules. The main goal of this phase is to increase the polarity of exogenous compounds [103].

The liver and small intestine (especially the jejunum and ileum) are the key organs responsible for various biotransformations leading to the formation of different conjugated forms of flavonoids; however, the kidney and other organs and tissues are also involved in flavonoid metabolism [24]. The most important metabolic reactions of flavonoid biotransformation are glucuronidation, sulphation, O-methylation, oxidation, reduction, and hydrolysis [103].

#### 5.2.4. The Most Important Metabolic Reactions in the Biotransformation of Polyphenols

The most important metabolic reaction to which flavonoids are subjected is the glucuronidation reaction. It consists of the transfer of glucuronic acid, which is bound to a specific substrate as UDP-glucuronic acid by the microsomal enzymes uridine-5′-diphosphate-glucuronisyltransferases (UGTs). The enzyme family UGT exhibits exceptional diversity in substrate recognition and catalyzes the glucuronidation of a large number of functional groups (e.g., -OH, -COOH, -NH_2_, and -SH). The reaction takes place on the luminal side of the endoplasmic reticulum, and UGT1A9 and UGT1A3, which are found in the intestine and liver, are thought to play the most important role [104]. The glucuronidation reaction allows the organism to make the substrates of the individual metabolic pathways more water-soluble. It is assumed that 80% of the metabolic pathways of flavonoids are due to the action of the enzyme UGT [105,106]. The chemical reaction of sulphation is mediated by sulphotransferases (SULT), enzymes responsible for catalyzing the sulphation of flavonoids and many drugs. In studies dealing with the sulphation of flavonoids, it was found that the sulphation process takes place mainly at the C7- OH position and that removal of the hydroxyl group reduces or inhibits the sulphation process [107,108,109].

Manach et al. [110] found that sulphate esters and glucuronides retain some of their antioxidant activity and still retard the in vitro oxidation of low-molecular-weight lipoproteins. However, Zhang et al. [111] have shown that glucuronidation of flavonoids reduces their biological activity. The glucuronides of daidzein and genistein have 10- and 40-fold lower affinity for estrogen receptors than the corresponding aglycones, respectively [112]. O-methylation of flavonoids is a common xenobiotic transformation in micro-organisms, plants, and mammals catalyzed by O-methyltransferases (OMTs) [113]. Compared to other flavonoids, flavan-3-ols are more susceptible to methylation in the jejunum, as evidenced by the specificity of catechol O-methyltransferase (COMT) for these compounds. O-methylation may reduce the biological activity of polyphenols: their antioxidant activity and effect on endothelial function [114].

Methylation may also affect the reduction of toxicity of flavonoids and polyphenols in general. Indeed, most polyphenols contain catechin groups that can be oxidized to toxic quinones in vivo. Similar quinones formed from endogenous estrogens and catecholamines lead to the formation of superoxide radicals through a reaction with nucleophilic molecules in cells. Even the smallest changes in the structure of flavonoids can significantly affect their activities in the body. Most often, biological activity is reduced and excretion accelerated, but there is also evidence of the formation of more biologically active compounds [115]. However, methylation of free hydroxyl groups in quercetin leads to more metabolically stable derivatives that readily penetrate membranes, resulting in better bioavailability. All these effects enhance the biological effects of quercetin, and the same applies to other flavones [115].

#### 5.2.5. Uptake of Polyphenol Metabolites into the Body

After the metabolism of polyphenols and their absorption in the small intestine, the metabolites of polyphenols enter the systemic circulation, where they are transported to the liver via the portal vein. The main metabolites found in the portal vein are mainly glucuronides and methylated glucuronides. These polar conjugates have also been found to cross the hepatocyte membrane and be further modified in other cell types, with most metabolites being excreted via the kidneys.

After the glucuronidation, sulphation, and methylation reactions, there are two types of metabolic pathways for polyphenol metabolites. One pathway leads to polyphenol metabolites in the plasma, which are then excreted into the urine via the kidneys, and the other pathway leads to transport into the colon.

A large proportion of ingested polyphenols reach the colon, including polyphenols that have not been absorbed in the small intestine (80–90%) and polyphenols that have been absorbed and metabolized (in the liver or small intestine) and then transported via membrane transporters or via bile directly back into the lumen of the colon. The colon contains a rich microbiota (10^12^ micrroorganisms/cm^3^) that has a high catalytic and hydrolytic capacity, leading to the degradation of polyphenols and the formation of a large number of new metabolites (second phase of metabolism) [116].

Enzymatic degradation of flavonoids in the colon leads to the formation of a large number of new metabolites because bacterial enzymes, unlike host enzymes, can catalyze many reactions, including hydrolysis, dehydroxylation, demethylation, ring cleavage and decarboxylation, and rapid deconjugation [115]. Enzymes of the intestinal microflora catalyze the degradation of the flavonoid scaffold into simpler molecules such as phenolic acids. The extent of absorption of flavonoid metabolites in the intestine is not yet fully understood. Therefore, it is necessary to determine the role of the gut microflora in the overall distribution and potential bioactivity of flavonoids in the diet [114,117,118,119].

### 5.3. The Relationship between the Intake of Food Rich in Polyphenols and the Change in Intestinal Microbiota

From today’s perspective, consumers of certain products demand that safe foods be used to feed animals. This is a challenge for today’s science and points to the use of natural food supplements in animal husbandry, with increasing attention being paid to the use of polyphenols. Polyphenols are not only known for their antioxidant effects, but they also benefit animals by improving their immunity. Phenols have anti-inflammatory and antimicrobial properties in the gut [120].

Multicellular organisms exist as metaorganisms consisting of both macroscopic hosts and symbiotic commensals [121]. The gut microbiota is a natural inhabitant of the gastrointestinal tract that resides in the host [122]. Its composition depends on the host’s homeostasis [123]. The host microbiota is a very living composition and is characterized by diversity and constant change over time due to the microbiological composition of the gastrointestinal tract. Changes in the microbiota also occur during breast milk consumption and during the transition to solid food. The absence of breast milk leads to changes in the gut microbiota, which may be the main cause of diarrhea [124]. The role of the gut microbiota is to digest impervious nutrients [125]. It is argued that polyphenols from foods, in general, influence the growth of probiotics and the gut microbiota in pigs and other animals, which could have an impact on gut health. It was originally claimed that food polyphenols alter the supply of biochemicals to the gut microbiome, which enables fermentation in the gut [126]. Changes in the distribution of certain bacterial species in the intestinal microbiome of the organism contribute to the antioxidant and anti-inflammatory effects, and further reduction of pathogenic species by polyphenols may stimulate the growth of probiotics [127,128]. It is assumed that polyphenols are most active in the intestines, which means that they can exert antioxidant, anti-inflammatory and antibiotic effects. At one point in the structure, hydrolyzed glycosidic compounds can react with enzymes of the gut microbiome in the form of aglycones before undergoing conjugation reactions [45,128]. Whole plants or their parts are a great asset when they are an integral part of the diet. Thus, plant leaf extracts inhibited human hepatocellular carcinoma (HepG2) cell growth [129] and induced cytotoxic activity against HepG2 cancer cells via oxidative stress in vitro [130], although the manifestation of activity will be markedly different under in vivo conditions due to the metabolism of polyphenols. Reference is also made to possible negative effects if polyphenols are consumed in excess and if they are taken alone and not as a limited supplement to the basic diet [131,132]. After ingestion, they are usually extensively metabolized in the gut microbiome of humans and animals before being absorbed into the bloodstream, where they are treated as xenobiotics and eventually excreted. Some polyphenols are known to interfere with the absorption of essential nutrients (e.g., iron and other metals formed from polyphenols), but their health benefits outweigh the potential negative effects [128,133,134]. Differences in the metabolism of polyphenols (i.e., their biotransformation and absorption) in the intestines may be due to mutual individual differences in the intestinal microflora and differences in the structures of polyphenols [135]. For example, many phytoestrogen flavonoids (e.g., genistein, apigenin, kaempferol, and naringenin) are more easily metabolized and absorbed in the form of aglycones, but the efficiency of their hydrolysis and absorption of glycosides (the most abundant form in plants) depends on the type of microflora [136,137]. The effects of polyphenol-rich food additives on the microbiome measured in various studies are listed in Table 2.

The individual sources of polyphenols used as supplements to animal feed in several studies are listed in Table 2. The microorganisms were determined from feces or in vivo, and the good bacteria whose concentration increased and the pathogens whose concentration decreased were listed. From the data presented, it can be concluded that polyphenols have a stimulating effect on the modulation (in a positive direction) of the intestinal microbiome of the animals tested.

## 6. Antioxidant and Pro-Oxidant Activity of Polyphenols

### 6.1. Antioxidant Activity of Polyphenols

Oxidative stress has become a major challenge for animal husbandry due to the impairment of animal growth. Discoveries about the intracellular antioxidant action of polyphenols as a plant additive in animal feed contributed significantly to the improvement of antioxidant activity. Due to the positive effect of polyphenols in preventing oxidative stress damage and eliminating excessively produced free radicals, polyphenols are receiving more and more attention [136].

The antioxidant effect of polyphenols in neutralizing free radicals involves the transfer of a hydrogen atom from the active hydroxyl group of the polyphenolic compound to a free radical (Ar-OH +R → Ar-O + RH) [113]. Di Meo et al. [135] have proposed four mechanisms for the reaction shown, which are illustrated in Figure 5.

In the first example, hydrogen atom transfer (HAT) occurs, a mechanism in which a hydrogen atom is transferred from an antioxidant (ArOH) to a radical (R·), reducing the radical (RH) and forming an antioxidant radical (AO). The mechanism identified with HAT in biological systems is called PCET (proton-coupled electron transfer) and is characterized by the pre-reaction complex in which the proton transfer occurs along the H-bond to one of the free O-atom pairs of the free radical.

The second mechanism involves the transfer of an electron (one-electron transfer-proton transfer, SET-PT), which occurs in two steps. In the first step the electron is transferred and the first radical cation reaction (ArOH·^+^) formed from antioxidants and anions (R^−^), takes place. In the second step, an exchange of protons takes place between the obtained products of the first reaction. 

In the third mechanism, proton loss by electron transfer takes place (Sequential Proton Loss-Electron Transfer, SPLET), in which two processes take place: the elimination of protons from antioxidants and the formation of anions (AO·) and electron transfer.

In the fourth mechanism (adduct formation), the radical forms a hydrogen bond with the HO group of the antioxidant. In a radiolytic solution, a radical that takes up an H ion can lead to the formation of a double bond in an aromatic ring.

### 6.2. Pro-oxidative Activity of Polyphenols

Polyphenols act as antioxidants at low concentrations, while at higher concentrations they have a pro-oxidant effect. This depends on many factors. One of the most important factors is the “seasonal type”, where weather conditions during the growing season of the plants, such as high air temperature and the amount of UV radiation, lead to the formation of higher concentrations of polyphenols in the plants. Therefore, the concentration of polyphenols is higher in food produced for animals during the growing season of plants, which may have a pro-oxidant effect. Polyphenols may have a pro-oxidative effect on other cells in environments where the partial pressure and concentration of oxygen are increased [136].

The same flavonoids can have both negative and positive effects, depending on the source and concentration of free radicals and the concentration of polyphenols in the administered dose [148]. The pro-oxidant properties of food polyphenols may deteriorate due to the lack of glutathione (GSH) in cells, the lack of chemical stability, and the activation of cellular copper ions (Cu^+^) [123] formed during autoxidation together with the semiquinone radical (oxidized flavonoid). The cytotoxic semiquinone radical reduces NADH, which is involved in the formation of ROS in redox reactions [149].

Copper ions contribute to the increase in superoxide radical concentration and the formation of hydrogen peroxide and hydroxyl ions. However, the pro-oxidative action of polyphenols may also have beneficial effects by inducing mild oxidative stress, generating antioxidants and xenobiotic-degrading enzymes, and contributing to a general cytoprotective effect [149,150]. Polyphenols are very susceptible to autoxidation [151].

The most studied property is the ability to protect the organism from free radicals and oxygenated reactive species formed during oxygen metabolism [152], which are also the main cause of radical damage in the cell [153]. Radical damage to cells leads to changes in net charge, which alters their osmotic pressure and leads to an increase in volume and death. Lipid peroxidation products derived from dead cells can also promote carcinogenesis [154]. Flavonoids prevent the damage caused by free radicals, in part by directly scavenging the radicals. Simply put, flavonoids are oxidized by radicals (R^−^) and converted into a more stable and less reactive form according to the following mechanism [155]:flavonoid (OH) + R^•^ → flavonoid (O^•^) + RH

The newly formed flavonoxy radical (O^−^) is stabilized by resonance, as the unpaired electron can and will be displaced throughout the aromatic ring, but it can also participate in reactions of dimerization, dismutation, recombination with other radicals, or oxidation to quinone, whether it reacts with radicals, other antioxidants, or perhaps biomolecules. The reaction of flavonoxy radicals with oxygen produces quinone and superoxide anions. This reaction is responsible for the undesirable pro-oxidant effect of flavonoids in healthy tissue, but may be helpful in combating tumor tissues. This confirms the assumption that the antioxidant capacity of flavonoids depends not only on the redox potential of the cell itself, i.e., the O^−^/OH potential, but also on the reactivity of the flavone oxyradical formed [152].

## 7. The Influence of Polyphenols on Animals

### 7.1. Effect of the Addition of Polyphenols on Animals

The addition of polyphenols to animal feed had no clear effect on animal growth. Inhibited secretion of digestive enzymes, increased protein excretion, and reduced digestibility of proteins and amino acids may have a detrimental effect on metabolism, as evidenced by a decrease in body weight and nutritional efficiency [152,153,154,155,156,157].

Fiesel et al. [100] reported a significant decrease in digestibility of total proteins and cellulose in weaners whose diets were supplemented with spent hops, a source of natural polyphenols. The supplement had no effect on performance and the gain-to-feed ratio was better in the experimental group than in the control group pigs (638 g/kg versus 579 g/kg).

The addition of Moringa oleifera, a source of quercetin and kaempferol, to the diet of chickens contributed to a significant increase in body weight at 21 days of age (low: 928 g; median: 932.5 g; high: 954.6 g) compared to chickens in positive and negative control groups (887.6 g; 918.7 g) and improved feed conversion (1.47; 1.44; 1.45) compared to 1.53 for positive control group during the experiment [158]. Supplementation of the chicken diet with 0.2% plant extracts of *Chelidonium majus*, *Lonicera japonica* and *Saposhnikovia divaricata* (sources of flavonoids, tannins, phenolic compounds, saponins, terpenoids and essential oils) resulted in a significant increase in final body weight (1949 g; 1930 g; 1930 g vs. 1845 g) and increased daily gain of about 3 g compared to the control chickens [143,159].

In a study by El-Iraqi et al. [160], chickens under heat stress (32–40 °C) were fed a diet enriched with dried peppermint and *Ginkgo biloba* (a source of flavonoids). This resulted in a significant increase in final body weight and a decrease in feed conversion compared to chickens whose diets were enriched with single herbs or vitamin C.

Viveros et al. [143] observed a significant decrease in body weight of 21-day-old chickens whose diets were enriched with grape seed extract (7.2 g/kg diet) compared to chickens in the other groups (486 g vs. 553; 557; 542 g). A significant reduction in feed conversion rate was also observed in chickens fed grape pomace concentrate (60 g/kg feed) or the antibiotic avoparcin (50 mg/kg feed) compared to chickens in the other groups (1.43; 1.43 vs. 1.51). The conversion rate of the feed mixture of pigs whose diet was enriched with polyphenols (from grape seeds and grape marc) increased compared to the control group (652 vs. 624 g/kg; *p* < 0.05, which is a significant correlation) [161,162]. In the study by Lipiński et al. (2015) [163], dietary supplementation with polyphenols (grape seeds and onions) had no effect on growth performance, percentage carcass weight loss, breast muscle yield, or meat composition in broilers.

In the study by Flis et al. [164], phenolic compounds in oat grains fed to pigs at the end of fattening (45% of the diet) had no effect on animal growth or carcass leanness. Dietary supplementation with cranberry extract, a rich source of phenolic compounds, had no effect on body weight or feed efficiency in poultry [165]. Simitzis et al. [166] showed that dietary supplementation with hesperidin and tocopherol acetate had no significant effects on growth or slaughter weight of broilers. The addition of grape by-products did not improve efficiency parameters, while herbs rich in flavonoids gave positive results, especially in animals under heat stress [4].

### 7.2. Immunomodulatory Effect of Polyphenols on Intestinal Health of Animals

The body’s immune system consists of innate and adaptive immunity. The first line of defense is innate immunity, which forms barriers between the organism and the environment [167]. The innate system can be divided into cellular and non-cellular systems. The cellular system consists of several subgroups of cells, namely, dendritic cells (DC), monocytes, macrophages, granulocytes, and natural killer (NK) cells. The non-specific systems are very diverse and range from mucosal barriers to complex protein molecules. The basic function is to prevent the invasion of pathogens and destroy them by phagocytosis [168]. The adaptive immune system consists of T and B cells. B cells secrete antibodies, while T cells synthesize cytokinins that destroy infected or malignant tissue. Many studies have confirmed the positive relationship between microflora and host health. The bioefficacy of polyphenols in interaction with the gut microbiome can promote the development of beneficial micro-organisms in the host gut and positively modulate the gut microbiome [169]. Previous studies in mice have confirmed that polyphenols positively modulate the gut microbiota by altering the composition of certain bacteria in the gut, such as *Bifidobacterium, Proteobacteria, Actinobacteria, Deferribacteres, Lactobacillus, Helicobacter, Desulfovibrio, Adlercreutzia, Prevotella*, and *Flexispira* [170]. For example, it was found that the addition of resveratrol to the diet of mice at a dose of 200 mg/kg per day improved the dysbiosis of the intestinal microbiota caused by a high-fat diet, i.e., it resulted in an inhibition of the growth of *Enterococcus faecalis* and an increase in the growth of *Lactobacillus and Bifidobacterium* [171]. Similar results and positive effects were obtained with the addition of blueberry polyphenol extract to the diet of mice, i.e., weight gain was prevented and fat metabolism returned to normal [172]. It was found that the addition of blueberry polyphenol extract modulated the composition of the microbiome—the populations of Proteobacteria and Deferribacteres increased and the population of Actinobacteria in the gut of mice decreased. Specifically, populations of *Bifidobacterium, Desulfovibrio, Adlercreutzia, Helicobacter*, and *Flexispira* were found to increase, and *Adlercreutzia* and *Prevotella* decreased, compared to mice fed a high-fat diet without polyphenols [167]. Experimental evidence suggests that polyphenols play an important role in the proliferation of beneficial gut bacteria in mice, such as *Akkermansia muciniphila*, and the reduction in pathogenic gut microbiota [168].

The interactions between the gut microbiota and polyphenols in humans are crucial as they modulate the gut microbiota and not only influence the composition of gut bacteria, but also improve the bioavailability of polyphenols through the formation of more bioactive metabolites, enhance their health-promoting effects, protect epithelial cells, and prevent inflammatory processes in the gut. Polyphenols suppress the development of pathogenic bacteria and promote the growth of beneficial bacteria that act as prebiotics [169]. The gut microbiota enhances the bioavailability of polyphenols by producing more bioactive metabolites that increase health effects and influence gut morphology [169]. The influence of polyphenols in stress during weaning of piglets and calves improves digestion and absorption of nutrients, improves the function of the intestinal barrier, improves the function of the intestinal microbiota, and thus provides positive effects [170]. The study of the concentration of polyphenols in body tissues is not directly related to their concentration in the diet. In pigs whose diets were supplemented with 50 mg/kg quercetin for 4 weeks, quercetin levels were higher in the kidneys (6.31 nmol/g) and colon (13.92 nmol/g) than in the liver (2.83 nmol/g) or plasma (0.67 mol/L). In contrast, quercetin concentrations in tissues remained unchanged (3.78 nmol/g in liver, 1.84 nmol/g in kidney), while they increased in plasma (1.1 mol/L) when pigs’ diets were supplemented with higher doses of quercetin (500 mg/kg) for 3 days [4]. This indicates that increased concentrations of polyphenols were found only in organs involved in the metabolism of polyphenolic compounds. Concentrations of unabsorbed dietary phenolic compounds have significant effects on the gut environment by suppressing or stimulating the growth of some gut microbiota. Dietary polyphenols have prebiotic properties and act antimicrobially by promoting the growth of certain strains of beneficial bacteria (*Bacillus* spp. and *Lactobacillus* spp.) in the digestive tract, while competitively excluding certain pathogenic bacteria and stabilizing the gut microbiota and strengthening the immune system of animals [171].

### 7.3. The Effectiveness of Polyphenols on the Quality of Products of Animal Origin

Feeding animals diets rich in n-3 PUFA has been shown to improve the nutritional quality of fats in animal products. However, it has been shown that a dietary strategy with increased PUFA content makes the products of these animals susceptible to lipoperoxidation. For this reason, the use of antioxidants in the diet has been recommended to limit lipid peroxidation, maintain animal health, and achieve better product quality [172]. Thus, vitamin E was resorted to as a synthetic antioxidant. However, high doses of vitamin E have been reported to have a pro-oxidant effect. Therefore, the use of a natural antioxidant has been recommended to optimize antioxidant protection in animals fed a PUFA-rich diet. Polyphenols are of interest among natural antioxidants. Various herbs, spices and plant residues are rich in polyphenols and could prevent lipid oxidation. Studies on poultry, for example, have confirmed that plant extracts or dietary hesperidin supplementation prevent lipoperoxidation [101,166,173].

From the detailed review by Serra et al. (2021) (Table 3) [4], it appears that the enrichment of animal feed with polyphenols improves the products of animal origin (based on meat or milk) without significant negative effects on animal health. Food-derived polyphenols have been shown to undergo no significant metabolic changes, thus enriching meat [174,175] and dairy products [176,177,178,179,180]. It has also been shown that the addition of 5% dried GP, a rich PUFA source, to the diet of pigs improves the concentration of total n-3 PUFA (especially α-linolenic acid) in muscle after 12 days of feeding [181]. Polyphenols have been found to contribute to the regeneration of tocopherol in broiler plasma due to the potential reduction of one electron, as well as protecting vitamin E from oxidation and thus slowing lipid oxidation [182].

The addition of polyphenols makes the animals’ diet more diverse, leading to greater diversity of gut flora and a better immune status for the animals. Combined with the ability for the animals to freely choose additional feed of plant origin containing polyphenols, this leads to a reduction in stress in the animals [183,184]. It has been shown that animals that are free to choose their feed optimise their individual macronutrient requirements, which has a positive effect on their health [183], whereas a monotonous and uniform diet causes stress [184]. Modern genetically bred pigs have not lost the ability to optimally select their feed in terms of optimising individual nutrient requirements [185]. The animals’ free choice of feed improves their lives, which contributes to a better health status of the animals.

In the future, more precise studies need to be defined that combine research on the addition of polyphenols to feed with the partial freedom of animals in intensive farming to choose the type and amount of feed [186]. In this way, a better quality of food of animal origin used for human consumption will be achieved. So there is a great opportunity to use natural antioxidants in the future to protect animal health and improve food of animal origin for human consumption.

## 8. Conclusions

Medicinal herbs and preparations made from medicinal herbs are one of the oldest ways of treating various diseases. Plants are known to contain many bioactive chemical compounds attributed to health effects. One of the most common secondary metabolites of plants are polyphenols. The process of absorption, metabolism, distribution and excretion from the organism is crucial for the determination of potentially bioactive effects in vivo and for the prevention of a number of diseases associated with oxidative stress. The bioavailability of polyphenols is related to the inter-reactions of polyphenols in the different phases of digestion. Most of the ingested polyphenols (90–95%) reach the large intestine. It is known that the intestinal microbiota is rich (it contains about 10^12^ microorganisms/cm^3^). Polyphenols are degraded by the microbiota and a large number of metabolites are formed. The intestinal microbiota improves the bioavailability of polyphenols, whereby metabolites are formed that affect the modulation of the intestinal microbiota in a positive direction, increasing the concentration of beneficial bacteria, while decreasing pathogenic ones. Previous studies on pigs and poultry have confirmed that plant polyphenols alleviate inflammatory conditions and improve intestinal microbiota. Polyphenols are known to have pronounced antioxidant activity in vitro, such as ROS removal, metal ion chelation, and an effect on increased transcription of antioxidant enzymes. However, due to low absorption, polyphenols are degraded in a xenobiotic manner, so it is necessary to compare them with other antioxidants such as vitamin C, vitamin E, albumins, uric acid and glutathione. It has also been confirmed that plant polyphenols can lower the level of ROS in animals with intestinal inflammation. Feeding pigs with grape pomace improved growth performance and changed the composition of fatty acids in finished products, which is important for human nutrition and such finished products. In addition to the positive effects, polyphenols also have a pro-oxidative effect and form complexes with metals, making them less available.

Further research and experimentation with polyphenolic compounds as feed additives for animals is a potential that is certainly worthy of much more research, as a relatively small number of experiments have provided positive evidence of an effect on the health status of animals. In addition to finding the right ratio of polyphenol preparations and plants in which they can best be used, and depending on the characteristics of the animal we want to improve, it is also important to focus research on maintaining the quality of polyphenol preparations so that they can be used as efficiently as possible. In order to achieve better overall results in terms of animal health, the quality of their lives and the quality of food of animal origin, the addition of polyphenols to the diet can be combined in research and in practice with the free choice approach to feed.

## Figures and Tables

**Figure 1 antioxidants-12-01141-f001:**
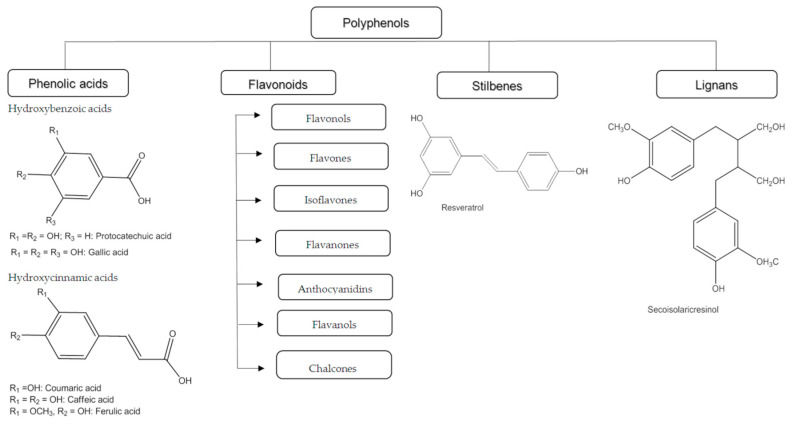
Classification of polyphenols based on their chemical structure (arranged according to Bešlo et al., 2022, [21]).

**Figure 2 antioxidants-12-01141-f002:**
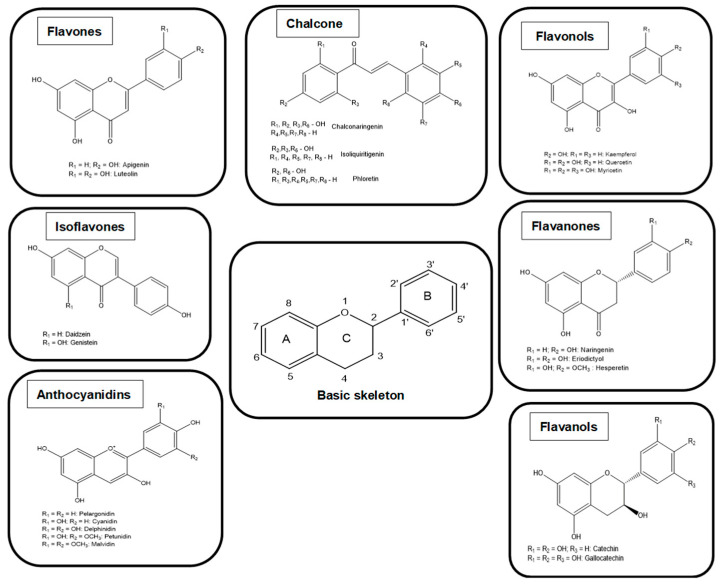
Chemical structures of flavonoids.

**Figure 3 antioxidants-12-01141-f003:**
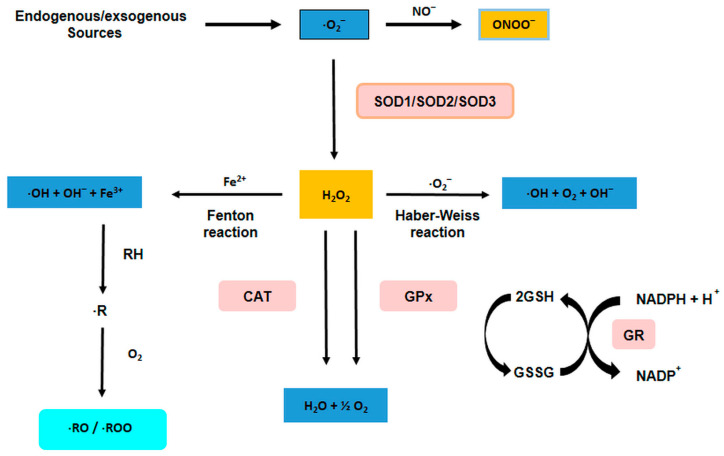
Sources and generation of reactive oxygen species (ROS). Abbreviations: CAT (catalase), GR (glutathione reductase), SOD (superoxide dismutase), RH (lipid membrane), R (alkyl radical).

**Figure 4 antioxidants-12-01141-f004:**
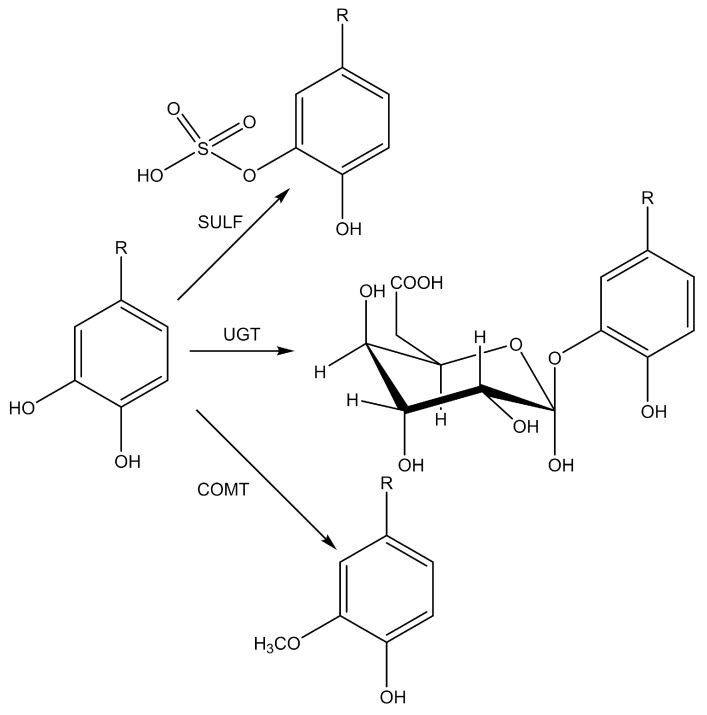
Conjugation reaction of phenolic hydroxy groups: glucuronidation, sulphonation, and/or methylation by uridine-5-diphosphate glucuronosyltransferase (UGT), sulphotransferase (SULT), and catechol-O-methyltransferase (COMT).

**Figure 5 antioxidants-12-01141-f005:**
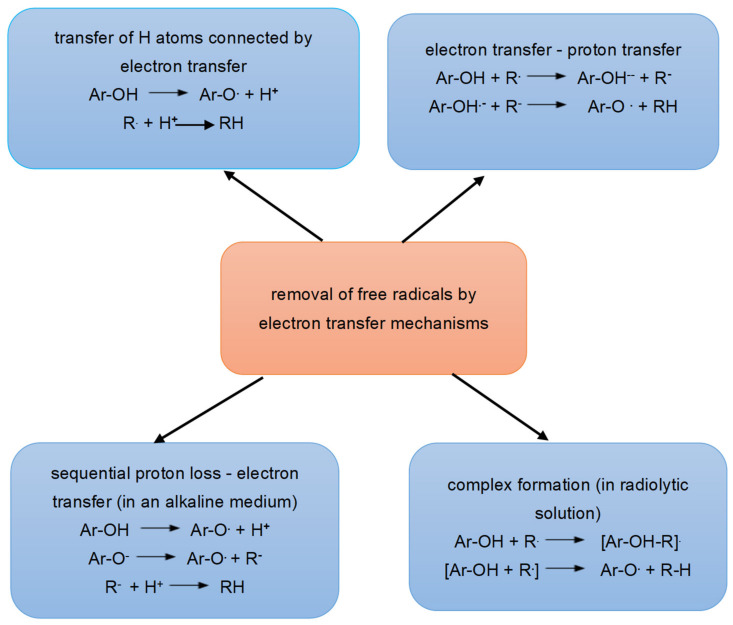
Polyphenol free radical scavenging activity by the hydrogen transfer mechanism. Abbreviations: R·—free radical, Ar-OH—phenolic compound.

**Table 1 antioxidants-12-01141-t001:** Reactive species (ROS and RNS) contributing to the occurrence of oxidative stress ^a^.

Name of the Substance		Symbol	Half-Life ^b^	Characteristic
**Radicals**	Superoxide	O_2_·	10^−6^ s	unstable molecule, signaling molecule
	Hydroxyl	OH·	10^−10^ s	very reactive and unstable species, it is created in the Fenton and Haber–Weiss reaction with an iron catalyst
	Alkoxyl radical	RO·	10^−6^ s	organic (lipid) radical
	Peroxyl Radical	ROO·	17 s	it is formed from organic hydroperoxide (ROOH), by removing hydrogen
	Nitric oxide	NO·	s	environmental toxin, endogenous signaling molecule
	Nitrogen dioxide	NO_2_·	s	highly reactive species, environmental toxin
**Non radical**	Hydrogen peroxide	H_2_O_2_	Stable	a cellular toxin, signaling role, generation of other free radicals
	Singlet oxygen	^1^O_2_	10^−6^ s	the first excited form of oxygen
	Ozone	O_3_	s	environmental toxin
	Organic peroxide	ROOH	Stabile	it easily decomposes into radicals, so it serves as a catalyst for radical reactions
	Peroxynitrous	ONOO^−^	Stabile	highly reactive species, environmental toxin
	nitrogen oxides	NO_x_	s	environmental toxin, including NO and NO_2_ derivatives formed in the combustion process

^a^ Arranged from references [71,72]. ^b^ The half-life of some radicals depends on the surrounding medium, for example, the half-life of NO· in an air-saturated solution can be a few minutes; s seconds.

**Table 2 antioxidants-12-01141-t002:** Presentation of the effect of polyphenols on intestinal microbiota.

Polyphenols and Source	Sample	Impact on Microbiome	References
**Red wine extract**	Fecal (human) in vitro	Increase: *Bifidobacterium* spp.	[138]
		*Lactobacillus/Enterococcus* spp.	
		*Bacteroides* spp.	
		Inhibit: *Clostridin hystolyticum*	
**Grape pomace**	Fecal (human) in vitro	*Increase: Bifidobacterium* spp.	[139]
		*Clostrida*	
		*Campylobacter*	
		*Inhibit: Escherichia coli*	
		*Salmonella*	
**Blueberry extract**	Fecal (human) in vitro	*Increase: Lactobacillus*	[140]
		*Bifidobacterium* spp.	
**Grape seed extract fraction**	Fecal (human) in vitro	Increase: *Lactobacillus/Enterococcus* spp.	[141]
		Inhibit: *Clostridin hystolyticum*	
**Tea polyphenols**	Fecal (pigs) in vitro	Increase: *Lactobacillus*	[142]
		Inhibit: *Bacteroidaceae*	
		*Clostridium perfringens*	
**Red wine polyphenols powder**	Fecal (rats) in vitro	Increase: *Lactobacilli*	[143]
		*Bifidobacteria*	
		Inhibit: *Propionibacteria*	
		*Bacteriodes, Clostridia*	
**Grape pomace concentrate, Grape seed extract**	Fecal (broiler chicks) in vitro	Increase: *E. Coli, Enterococcus* spp., *Lactobacillus* spp.	[144]
**Lowbush wild blueberries**	Fecal (rats) in vitro	Incerase: *Thermonospora* spp., *Corynebacteria spp*.	[145]
		*Slackia* spp.	
		Inhibit: *Lactobacillus* spp. *Enterococcus* spp.	
**Apple pomace**	In vivo (rats)	*Increase: Lactobacillus*	[146]
		*Bifidobacterium*	
		*Bacteriodoceae*	
		*Eubacterium*	
		Inhibit:	
		*Bacteroides* spp.	
**Tannin supplementation**	In vivo (mouse)	Increase: *Bacteroides, Lactobacillus*	[147]
		Inhibit: *Clostridium*	

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
