# Peer review of "Antioxidant Activity, Metabolism, and Bioavailability of Polyphenols in the Diet of Animals"

_antioxidants, 2023, doi:10.3390/antiox12061141_

Round 1

Reviewer 1 Report

Ms. Ref. No.:  Antioxidants 2362565

The use of polyphenols in the diet of monogastric animals

The work aims to address the issue of supplementation of monogastric animals with polyphenols, indicating its effect on health. However, there is only one specific section in which a review is made on this subject, the subsection 6.3. Effect of the addition of polyphenols on animal growth. The work does not present any originality in this regard and is a compendium of already published information. Therefore, this paper lacks originality and the title does not correspond to the content.

The manuscript needs to be rearranged and can be accepted with major revision.

1.    In the title:  should be changed the title should be changed since only slight mentions are made of the metabolism of polyphenols in monogastric animals. The title could indicate the antioxidant effect of phenolic compounds, also mentioning their metabolism and bioavailability.

2.    In the Introduction section: In line 72 “Animals include horses, pigs, dogs, and rabbits”. Birds should be included in this phrase. This is necessary since the revision refers to chickens.

3.    In the section 3. Oxidative stress and defense against oxidative stress subsections 3.3 and 3.4 should be addressed first, since in the title the oxidative stress is indicated first. This section needs to be rearranged

4.    In 5.1. Digestion/metabolism/biotransformation and bioavailability of polyphenols in monogastric animals subsection: not the slightest mention is made of monogastric animals. Therefore the title of subsection must be changed, eliminating monogastric animals.

5.    In the subsection 5.3. The relationship between the state of the intestinal microflora and polyphenols. In the title, microflora should be changed to microbiota.

6.    The legend of figure 5 is repeated but with two different texts.

7.    The subsection 6.3. Effect of the addition of polyphenols on animal growth it is the most innovative section and the one that would fit the title. Therefore it should be more widely developed.

Author Response

Ad. 1. The title was changed to “Antioxidant activity, metabolism, and bioavailability of polyphenols in the diet of animals”.

Ad. 2. The introduction has been corrected and slightly expanded.

Ad. 3. The suggestion is accepted - and in section 3, the order has been changed so that former 3.3 is now 3.1, and former 3.4. is now 3.2, while the former 3.1 and 3.2 are now 3.3, and 3.4, respectively.

Ad. 4. The title of Subsection 5.3. is changed, i. e. ‘monogastric’ is deleted

Ad. 5. The title of subsection 5.3. is now changed to: “The relationship between the intake of food reach in polyphenols and the change of intestinal microbiota”

Ad. 6. The second legend was deleted in Figure 5

Ad 7. The former subsection 6.3. is now 7.1. entitled "Effect of the addition of polyphenols on animals". In addition, subsection 7.2. ("Immunomodulatory effect of polyphenols and intestinal health of animals") is added, as well as 7.3. "The effectiveness of polyphenols on the quality of products of animal origin".

Reviewer 2 Report

This review deals with the antioxidant properties of polyphenols in monogastric species.

The paper is well written and references appear to be adequate to give a complete picture of the present knowledge on the topic. Thus, it represents a significant contribution to the field and is worth of publication.

Minor revision to improve the scientific soundness of  the review should be made before it can be accepted for publication:

In the abstract, the possible implication of the polyphenols on the improvement of the quality of animal origin products has been mentioned (in particular on the PUFA content), by contrary, in the main text no information concerning this aspect was reported. Please add a brief paragraph on the effects of polyphenols on the quality of animal origin products.

You only focused on the growth promoter function of polyphenols. This is ok, but others effects have  been described in the literature (health-promoting properties such as cardiovascular effects, a positive cancer, the effects on animal products…) The discussions must be improved accordingly.

Thus, the abstract must be remodulated including the different effects of polyphenols in monogastric species.

I also suggest to add a table with the different use of polyphenols (with the related references) in monogastric nutrition.

In the paragraph 6.3. Effect of the addition of polyphenols on animal growth, the effects of the inclusion of different feedstuffs already known for their polyphenol content (peppermint, ginkgo biloba, grape seed….) have been reported. Despite that, no data concerning which kind and/or the quantification of these compounds are reported. Therefore, the potential effects of these studies need to be detailed.

The conclusion must be rewritten according to the improvement of the discussion.

Author Response

Ad. 1. The PUFA has now been amended both in summary and in separate subsection 7.3. The effects of polyphenols on the quality of products of animal origin

Ad. 2. In addition to the described effects of polyphenols on growth, other characteristics have also been described in the revised manuscript, such as intestinal inflammation.

Ad.3. The discussion has been revised and rearranged in accordance with the changes in the manuscript.

Ad. 4. The Conclusion has been revised and improved.

Ad. 5. Table 2.was added (“Table 2.  Presentation of the effect of polyphenols on intestinal microbiota “) to subsection 5.3. “The relationship between the intake of food reach in polyphenols and the change of intestinal microbiota”.

Ad. 6. Sub-chapter 6.3. is now 7.1. “Effect of the addition of polyphenols on animals” and new sub-chapter 7.2. is added entitled “Immunomodulatory effect of polyphenols and intestinal health of animals”, as well as 7.3. “The effectiveness of polyphenols on the quality of products of animal origin”

Ad. 7. Conclusions have been revised and adapted to the manuscript

Reviewer 3 Report

The paper is in the form of a review for a topic on which there is much bibliography and depth studies both on individual chemical compounds in their entire chain and in their application aspects in feedstuffs aimed at livestock and companion animals.

Based on the premise the paper is presented in an unclear form, the monogastric animals mentioned ( line 72)include pigs, horses, dogs and rabbits.

These animals should also be classified according to their use for example horses intended for human food or not, pigs is generic and unrelated to the classification used for breeding stages and related feeding needs.

For dogs the issue is still different resulting in health protection requirements.

For the parts related to metabolism and biotransformation, we do not highlight any particularities that distinguish the behavior of polyphenols in different animals.

6.3 The authors make several references to chickens.

The bibliography cited and consulted although widely represented is not discussed in an organized and structured form, there is no evidence of adequate design of the work by the authors such that the same has adequate relevance.

Author Response

Ad. 1. 'Monogastric' animals is deleted from the title of the manuscript, and the title is changed - as mentioned above – to: “Antioxidant activity, metabolism, and bioavailability of polyphenols in the diet of animals”.

Ad. 2. The effects of polyphenols on individual animals are described in sub-sections 7.1. and 7.2.

Ad. 3. The manuscript has now undergone a major revision, and individual parts and sub-chapters have been added.

Ad. 4. The conclusion has been rearranged, improved and adapted to the revised manuscript.

Round 2

Reviewer 1 Report

The authors have taken into account most of the suggestions and have expanded and developed those sections that are most innovative. Therefore I consider that the manuscript should be published.

Reviewer 3 Report

The revision made by the authors resulted in an appreciable change, the work was well completed and is aligned with the requirements.